# Divergent Pharmacology and Biased Signaling of the Four Melanocortin-4 Receptor Isoforms in Rainbow Trout (*Oncorhynchus mykiss*)

**DOI:** 10.3390/biom13081248

**Published:** 2023-08-16

**Authors:** Ren-Lei Ji, Ting Liu, Zhi-Shuai Hou, Hai-Shen Wen, Ya-Xiong Tao

**Affiliations:** 1Department of Anatomy, Physiology and Pharmacology, College of Veterinary Medicine, Auburn University, Auburn, AL 36849, USA; rlj0027@auburn.edu (R.-L.J.); tzl0057@auburn.edu (T.L.); 2The Key Laboratory of Mariculture, Ministry of Education, Ocean University of China, Qingdao 266100, China; wenhaishen@ouc.edu.cn

**Keywords:** rainbow trout, MC4R, constitutive activity, biased signaling, allosteric ligand, paralogs

## Abstract

The melanocortin-4 receptor (MC4R) is essential for the modulation of energy balance and reproduction in both fish and mammals. Rainbow trout (*Oncorhynchus mykiss*) has been extensively studied in various fields and provides a unique opportunity to investigate divergent physiological roles of paralogues. Herein we identified four trout *mc4r* (*mc4ra1*, *mc4ra2*, *mc4rb1*, and *mc4rb2*) genes. Four trout Mc4rs (omMc4rs) were homologous to those of teleost and mammalian MC4Rs. Multiple sequence alignments, a phylogenetic tree, chromosomal synteny analyses, and pharmacological studies showed that trout *mc4r* genes may have undergone different evolutionary processes. All four trout Mc4rs bound to two peptide agonists and elevated intracellular cAMP levels dose-dependently. High basal cAMP levels were observed at two omMc4rs, which were decreased by Agouti-related peptide. Only omMc4rb2 was constitutively active in the ERK1/2 signaling pathway. Ipsen 5i, ML00253764, and MCL0020 were biased allosteric modulators of omMc4rb1 with selective activation upon ERK1/2 signaling. ML00253764 behaved as an allosteric agonist in Gs-cAMP signaling of omMc4rb2. This study will lay the foundation for future physiological studies of various *mc4r* paralogs and reveal the evolution of MC4R in vertebrates.

## 1. Introduction

Melanocortin-4 receptor (MC4R), a Family A rhodopsin-like G protein-coupled receptor (GPCR), primarily couples to the stimulatory heterotrimeric G protein and activates adenylyl cyclase, resulting in elevated intracellular cyclic adenosine monophosphate (cAMP) generation to regulate downstream signaling [1]. α-Melanocyte-stimulating hormone (α-MSH) and other peptides derived from posttranslational processing of proopiomelanocortin (POMC) are endogenous agonists, and agouti-related peptide (AgRP) is an endogenous inverse agonist of MC4R. MC4R has high expression in the central nervous system and is essential for regulation of energy homeostasis, as well as reproduction and sexual function in mammals [1,2,3]. *Mc4r*-deficient mice show an obese phenotype with increased food intake and decreased energy expenditure [4,5]. More than 300 mutations in human *MC4R* have been identified from patients, and *MC4R* mutations are considered to be the leading cause of monogenic obesity [3,6,7,8,9].

Mc4r has also been studied in teleosts. Fish *mc4r* is widely distributed in both the central nervous system and the periphery [10,11,12,13,14,15,16,17,18,19]. Activation of Mc4r with agonists decreases food intake in rainbow trout and goldfish, while injection of antagonists results in increased food intake [11,20,21,22]. Aspiras et al. reported that Mexican cavefish *mc4r* mutations contribute to elevated appetite, starvation resistance, and growth, playing crucial roles in adaptation to a nutrient-poor environment [23]. Furthermore, Mc4r is shown to play key roles in sexual behavior and reproduction in teleosts [24,25,26,27] (reviewed in [9]). These studies indicate that teleost Mc4r is also an important regulator in energy balance and reproduction. Hence, a better understanding of physiological functions of teleost Mc4r will be beneficial for growth and artificial breeding of cultured fish.

Human MC4R (hMC4R) shows modest constitutive activity, which is crucial for modulating energy homeostasis [28,29]. The loss of basal activity in *MC4R* mutations is considered to be one cause of obesity [30,31,32]. Interestingly, high constitutive activity has been documented in Mc4r from several fishes, including zebrafish [33], spotted scat [13], grass carp [14], swamp eel [15], orange-spotted grouper [16], sea bass [34], topmouth culter [18], and snakehead [19]. The potential relevance of high basal activity in teleost Mc4r should be further investigated. Mutants of cavefish Mc4r with reduced maximal response and basal activity contribute to enhanced appetite and growth [23]. Additionally, melanocortin-2 receptor accessory protein 2- and/or AgRP-mediated suppression of Mc4r is essential for promoting zebrafish and topmouth culter growth [18,35,36,37]. Thus, constitutive activity may play a vital role in the modulation of energy balance in fish, and decreasing the constitutive activity of teleost Mc4r may provide new strategies for food intake and growth promotion in aquaculture. 

Biased signaling or functional selectivity, the preferred activation of one signaling pathway over another, has been reported in numerous GPCRs and utilized in drug design for GPCRs to develop therapeutics with decreased side effects [38,39,40,41,42,43]. In addition to Gαs-cAMP signaling, hMC4R can also activate the extracellular signal-regulated kinases 1 and 2 (ERK1/2) [44,45,46,47]. Biased agonism has also been reported with hMC4R [43,48,49,50]. However, only one study has investigated the biased signaling of fish Mc4r so far [51].

Rainbow trout (*Oncorhynchus mykiss*), cultured worldwide, belongs to the salmonid family and has been extensively studied in various fields, such as toxicology, carcinogenesis, ecology, immunology, physiology, and nutrition [52,53]. The presence of whole-genome duplication of rainbow trout also provides a unique opportunity to explore the early evolutionary feature of a duplicated vertebrate genome and contributes to investigating the divergent physiological roles of the paralogues [54,55]. 

In this study, we identified four paralogs of *mc4r* genes in rainbow trout, and we further investigated the pharmacological characteristics of the four trout Mc4r isoforms (omMc4rs) encoded by these duplicated genes. Inverse agonists have been extensively used in mammalian MC4Rs. However, the effects of mammalian MC4R inverse agonists on teleost Mc4r are not well understood [51]. Especially, small molecule inverse agonists, which could potentially be added into the feed, will contribute to developing novel feeding strategies in aquaculture. Thus, several MC4R inverse agonists, such as AgRP, MCL0020 [56], ML00253764 [57], and Ipsen 5i [58], were used to explore their effects on two signaling properties (Gαs-cAMP and ERK1/2 signaling) of the four trout Mc4rs. This study will facilitate the investigation of potential physiological functions of different *mc4r* paralogs, as well as contribute to revealing the evolution of MC4R in vertebrates.

## 2. Materials and Methods

### 2.1. Ligands and Plasmids

[Nle^4^, D-Phe^7^]-α-MSH (NDP-MSH) was obtained from Peptides International (Louisville, KY, USA). Human α-MSH was purchased from Pi Proteomics (Huntsville, AL, USA). Human ACTH (1–24) and AgRP (83–132) were obtained from Phoenix Pharmaceuticals (Burlingame, CA, USA). Ipsen 5i and 2-[2-[2-(5-bromo-2-methoxyphenyl)-ethyl]-3-fluorophenyl]-4,5-dihydro-1H-imidazol (ML00253764) were synthesized by Enzo Life Science (Plymouth Meeting, PA, USA). Ac-D-2-Nal-Arg-2-Nal-NH_2_ (MCL0020) was obtained from Tocris Bioscience (Ellisville, MO, USA). Two *pomc* genes, *pomca* and *pomcb*, were identified in the rainbow trout genome [53]. The trout α-Msh and Acth produced from Pomca had 100% and 92% similarities with human counterparts, respectively, and the α-Msh and Acth derived from trout Pomcb shared 92% and 83% identities with human counterparts, respectively (Appendix A). [^125^I]-cAMP and [^125^I]-NDP-MSH were iodinated using the chloramine T method [59,60]. The N-terminal myc-tagged human MC4R (hMC4R) subcloned into the pcDNA3.1 vector was generated as previously described [61]. N-terminal myc-tagged trout Mc4r plasmids were commercially synthesized and subcloned into pcDNA3.1 by GenScript (Piscataway, NJ, USA).

### 2.2. Homology, Phylogenetic, and Chromosome Synteny Analyses 

Four rainbow trout *mc4r* gene sequences were obtained from the National Center for Biotechnology Information (NCBI) database (LOC110535569, LOC110490013, LOC110508705, and LOC110530343). Multiple sequence alignments of different MC4Rs were performed with Clustal Omega (EMBL’s European Bioinformatics Institute, Cambridge, United Kingdom) (https://www.ebi.ac.uk/Tools/msa/clustalo/) (accessed on 15 April 2022). The similarity among amino acid sequences of MC4Rs was calculated with NCBI genome browser (National Library of Medicine, Bethesda, MD, USA) (https://blast.ncbi.nlm.nih.gov/Blast.cgi) (accessed on 15 October 2021). A phylogenetic tree was performed via Mega 5.0 software (developed by Sudhir Kumar and Koichiro Tamura from the Center for Evolutionary Functional Genomics at Arizona State University, Tempe, AZ, USA, along with Masatoshi Nei from the Department of Biology at Pennsylvania State University, University Park, PA, USA) using neighbor-joining methods. Chromosome synteny was determined by the NCBI and Ensembl genome browser (EMBL’s European Bioinformatics Institute, Cambridge, United Kingdom) (https://useast.ensembl.org/index.html) (accessed on 8 November 2021). 

### 2.3. Cell Culture and Transfection

Human Embryonic Kidney (HEK) 293T cells, provided by American Type Culture Collection (Manassas, VA, USA), were cultured as described previously [18,62]. Cells were plated into 24-well or 6-well plates pre-coated with 0.1% gelatin and transfected with plasmids by the calcium phosphate precipitation method [63]. 

### 2.4. Flow Cytometry Assay

Flow cytometry was used to determine the cell-surface and total expression levels of trout and human MC4Rs as described earlier [34,64]. Cells were plated into 6-well plates and transiently transfected with empty vector and trout and human MC4R plasmids. Forty-eight hours after transfection, cells were washed twice with cold PBS. Cells were detached with 1 mL PBS by pipetting up and down. Two wells were collected for non-permeabilized cells and four wells for permeabilized cells. For permeabilized cells, cells were fixed with 200 μL of 4% paraformaldehyde and were permeabilized with 200 μL of 0.1% Triton 100. Non-permeabilized and permeabilized cells were incubated with primary antibody (mouse anti-myc 9E10 antibody) and secondary antibody (Alexa Fluro@488 goat anti-mouse antibody). The CytoFLEX LX flow cytometer (Beckman Coulter, Indianapolis, IN, USA) was used for analysis. Empty vector (pcDNA3.1) fluorescence was used for correction of background staining. The expression of trout Mc4r was calculated as the percentage of hMC4R, after subtraction of background staining [65].

### 2.5. Ligand-Binding Assay

A binding assay was conducted as described previously [61,66]. The ligands and final concentrations were as follows: α-MSH (from 10^−10^ to 10^−5^ M), ACTH (1–24) (from 10^−10^ to 10^−5^ M), AgRP (83–132) (from 10^−12^ to 10^−7^ M), MCL0020 (from 10^−11^ to 10^−6^ M), ML00253764 (from 10^−10^ to 10^−5^ M), and Ipsen 5i (from 10^−11^ to 10^−6^ M). 

### 2.6. cAMP Assay

Intracellular cAMP levels were determined according to previous protocol [59,61]. Two ligands, α-MSH and ACTH1–24 (from 10^−10^ to 10^−6^ M), were used in this study. To explore the effects of MC4R inverse agonists on the basal cAMP levels of trout Mc4r, omMC4Rs-transfected cells were treated with 10 nM AgRP, 1 μM MCL0020, 1 μM ML00253764, or 1 μM Ipsen 5i for 1 h before intracellular cAMP samples were collected and measured. 

### 2.7. ERK1/2 Phosphorylation Assay

HEK293T cells were transfected with omMC4Rs. Twenty-four hours after transfection, cells were washed twice and starved in media without serum for 24 h. Cells were then stimulated with either buffer alone or different compounds (1 μM α-MSH, 10 nM AgRP, 1 μM MCL0020, 1 μM Ipsen 5i, or 1 μM ML00253764) for 5 min as described previously [60,67]. Rabbit anti-pERK1/2 antibody (Catalog #4370, Cell Signaling, Beverly, MA, USA) and mouse anti-β-tubulin antibody (Developmental Studies Hybridoma Bank, University of Iowa, Iowa City, IA, USA) were used in this study. The bound antibodies were quantified using ImageJ 1.44 software (National Institute of Health, Bethesda, MD, USA). 

### 2.8. Statistical Analysis

Results were represented as mean ± SEM. GraphPad Prism 8.3 software (San Diego, CA, USA) was used to analyze data. The significance of differences was determined by one-way ANOVA.

## 3. Results

### 3.1. Nucleotide and Deduced Amino Acid Sequences of Trout Mc4rs

Four trout *mc4r* genes, including *mc4ra1*, *mc4ra2*, *mc4rb1*, and *mc4rb2* (GenBank: XM_021613317.1, XM_021589432.2, XM_021620635.1, and XM_021563090.1, respectively), were identified. Trout *mc4ra1* had a 1002 bp open reading frame (ORF) and encoded a putative protein of 333 amino acids (AAs) with a 37.44 kDa molecular mass (Appendix A); *mc4ra2* contained an ORF of 1014 bp that encoded a putative protein of 337 AAs (Appendix A); *mc4rb1* had a 1020 bp ORF with 339 AAs (Appendix A); *mc4rb2* encoded a putative protein of 337 AAs with a 1014 bp ORF (Appendix A). Each of the four trout Mc4rs had seven putative hydrophobic transmembrane domains (TMDs) with an extracellular amino terminus, three extracellular loops (ECLs), three intracellular loops (ICLs), and an intracellular carboxyl terminus (Figure 1). The predicted AA sequences in the TMDs of trout Mc4rs were highly conserved with those of other species. Several motifs, including PMY, DRY, and DPxxY motifs, were predicted at homologous positions with other MC4Rs (Figure 1). Two potential *N*-linked glycosylation sites in N-termini were observed in Mc4ra1 (Asn^2^ and Asn^16^), Mc4ra2 (Asn^2^ and Asn^16^), and Mc4rb1 (Asn^3^ and Asn^18^); Mc4rb2 had three potential *N*-linked glycosylation sites (Asn^2^, Asn^17^, and Asn^37^) (Figure 1 and Appendix A). In addition, potential protein kinase C phosphorylation sites in C-termini were observed in four trout Mc4rs (Figure 1 and Appendix A). 

Multiple sequence alignment analysis showed that omMc4ra1 shared high identities with omMc4ra2 (94%), salmon Mc4ra1 (99%), and salmon Mc4ra2 (90%), and lower identities with omMc4rb1 (78%), omMc4rb2 (77%), salmon Mc4rb1 (78%), and Mc4rb2 (77%). The omMc4ra2 shared the highest identity with salmon Mc4ra2 (99%); omMc4rb1 had 91% homology with omMc4rb2 and salmon Mc4rb2, and 99% homology with salmon Mc4rb1; omMc4rb2 had the highest homology with salmon Mc4rb2 (96%) (Figure 2). 

### 3.2. Chromosome Synteny and Phylogenetic Analyses of Trout Mc4rs

The four trout *mc4r* genes were located on different chromosomes: chromosome 8 (*mc4ra1*), chromosome 28 (*mc4ra2*), chromosome 11 (*mc4rb1*), and chromosome 15 (*mc4rb2*) (Figure 3). Chromosome synteny analyses revealed that trout *mc4ra1* and *mc4ra2* neighbors showed more highly conserved synteny to zebrafish and salmon *mc4ra1* and *mc4ra2*; trout *mc4rb1* and *mc4rb2* were orthologous to those of medaka, chicken, and human *MC4R* (Figure 3).

Phylogenetic tree analysis between omMc4rs and other MC4Rs showed that four omMc4rs were clustered into four different groups (Figure 4). Trout Mc4ra1 was nested with other Mc4ra1s of salmonids and trout, and similar results were observed in Mc4ra2, Mc4rb1, and Mc4rb2. In addition, omMc4ra1 was close to omMc4ra2 (Figure 4).

### 3.3. Cell-Surface and Total Expression of omMc4rs 

Flow cytometry was used to determine cell-surface and total expression of omMc4rs (Figure 5). Results showed that all four trout Mc4rs had higher cell-surface expression than hMC4R (Figure 5A). Trout Mc4rb1 also showed higher total expression compared to hMC4R (Figure 5B). Additionally, three trout Mc4rs, including omMc4ra1, omMc4ra2, and omMc4rb1, showed higher surface ratios (surface expression/total expression) at greater than 70%, and hMC4R only had around a 40% surface ratio (Figure 5C).

### 3.4. Ligand-Binding Properties of omMc4rs

Binding assays were used to investigate the binding properties of omMc4rs. Different concentrations of two unlabeled agonists, namely, α-MSH and ACTH (1–24), and four antagonists, namely, AgRP (83–132), ML00253764, Ipsen 5i, and MCL0020, were used as competitors with a fixed amount of ^125^I-NDP-MSH. The maximal binding values (B_max_) of four trout Mc4rs were 15.51 ± 3.62, 10.80 ± 0.49, 39.66 ± 2.06, and 14.32 ± 1.25% of that of hMC4R, respectively (Figure 6 and Table 1). Of note, omMc4rb2 had lower IC_50_s to α-MSH and ACTH, while omMc4ra2 showed higher IC_50_ to ACTH compared to hMC4R (Figure 6 and Table 1).

For the antagonists, only AgRP (83–132) could dose-dependently displace the ^125^I-NDP-MSH binding to the four omMc4rs with lower IC_50_ values than that of hMC4R (Figure 6 and Table 1). MCL0020 only displaced ^125^I-NDP-MSH binding to hMC4R and omMc4rb1, and omMc4rb1 showed a higher IC_50_ than that of hMC4R (Figure 6 and Table 1). ML00253764 and Ipsen 5i displaced ^125^I-NDP-MSH binding to hMC4R, with IC_50_ values of 1644.57 ± 215.24 nM and 399.86 ± 122.11 nM, respectively, and these two compounds could not displace ^125^I-NDP-MSH binding to the four trout Mc4rs (Figure 6 and Table 1).

### 3.5. Signaling Properties of omMc4rs

Both α-MSH and ACTH (1–24) were able to stimulate omMc4rs and dose-dependently increased intracellular cAMP generation (Figure 7 and Table 2). Trout Mc4rs showed lower maximal responses (R_max_) in response to α-MSH and ACTH (1–24) stimulation compared with hMC4R (Figure 7 and Table 2). Trout Mc4rb1 and Mc4rb2 had lower EC_50_s in response to α-MSH and ACTH (1–24) than that of hMC4R, whereas omMc4ra1 and omMc4ra2 showed higher EC_50_ values than hMC4R (Figure 7 and Table 2).

In this study, the basal signaling of omMc4ra1, omMc4ra2, omMc4rb1, and omMc4rb2, was 1.16-, 2.53-, 6.48-, and 0.83-fold that of hMC4R, respectively, indicating that trout Mc4rs might be constitutively active, especially omMc4ra2 and omMc4rb1 (Figure 8A).

### 3.6. Constitutive Activities of omMc4rs in Response to Four Antagonists

The basal activity of hMC4R and constitutively active mutant hMC4Rs were shown to be reduced by inverse agonists, such as AgRP (83–132), ML00253764, MCL0020, and Ipsen 5i [1,28,48,51]. To investigate their effects on the basal activities of trout Mc4rs, cells transfected with the four trout Mc4rs were treated with 10 nM AgRP (83–132), 1 μM MCL0020, 1 μM ML00253764, or 1 μM Ipsen 5i, respectively. Results showed that AgRP (83–132) acted as an inverse agonist for the four trout Mc4rs (Figure 8). ML00253764 acted as an agonist for omMcrb2 and significantly increasing the cAMP levels (Figure 8). Ipsen 5i and MCL0020 had no significant effect on the cAMP levels of the four trout Mc4rs (Figure 8).

### 3.7. ERK1/2 Signaling Properties of omMc4rs in Response to Five Ligands

To assess the potential of five ligands (α-MSH, AgRP, ML00253764, MCL0020, and Ipsen 5i) for modulation of ERK1/2 phosphorylation of trout Mc4rs, both basal and ligand-induced ERK1/2 signaling were investigated using Western blotting (Figure 9). Results showed that only omMc4rb2 had higher basal pERK1/2 levels compared to the empty vector, and the other three omMc4rs had similar basal pERK1/2 levels as the empty vector (Figure 9A,B). For ligand-induced signaling, α-MSH was shown to increase the ERK1/2 phosphorylation of four trout Mc4rs (Figure 9A,C–F). The pERK1/2 level in omMc4rb1-transfected HEK293T cells was significantly increased upon treatment with 10 nM AgRP (83–132), 1 μM ML00253764, 1 μM Ipsen 5i, or 1 μM MCL0020 (Figure 9A,E). The pERK1/2 of omMc4ra1 was only increased by AgRP (83–132), but not by ML00253764, Ipsen 5i, or MCL0020 (Figure 9A,C). Four ligands (AgRP (83–132), ML00253764, MCL0020, and Ipsen 5i) had no effect on ERK1/2 signaling of omMc4ra2 and omMc4rb2 (Figure 9A,D,F).

## 4. Discussion

In the present study, we identified four rainbow trout *mc4r* genes (*mc4ra1*, *mc4ra2*, *mc4rb1*, and *mc4rb2*). The four trout Mc4rs had similar structural features as the MC4Rs of other species (Figure 1). These conserved motifs and residues have been shown to play crucial roles in maintaining receptor structure and function [1]. Multiple sequence alignment, a phylogenetic tree, and chromosomal synteny analysis indicated that *mc4ra1* and *mc4ra2* might be two copies of *mc4ra*, and *mc4rb* may have two copies: *mc4rb1* and *mc4rb2* (Figure 1, Figure 2, Figure 3 and Figure 4). These results were possibly due to whole-genome duplication, similar to other salmonids, indicating that *mc4r* may have undergone different evolutionary processes. In Atlantic salmon, four *mc4r* (a1, a2, b1, and b2) were identified and have high expression in the different regions of brain [68]. Administration of MC4R agonist decreases food intake, and antagonists result in increased food intake in rainbow trout [22]. These indicated that four different Mc4rs of trout and salmon might have various roles in appetite regulation and energy balance.

To further investigate the pharmacology of omMc4rs, ligand binding and cAMP assays were performed using hMC4R for comparison. Similar to previous studies in spotted scat [13], grass carp [14], swamp eel [15], sea bass [34], orange-spotted grouper [16], topmouth culter [18], and snakehead [19], four omMc4rs showed lower binding capacity than hMC4R (Figure 6 and Table 1), while four omMc4rs had higher cell-surface expression than that of hMC4R (Figure 5), which was different than sea bass Mc4r, which had lower cell-surface expression [34]. Additionally, omMc4rb2 had lower IC_50_ values than α-MSH and ACTH, while omMc4ra2 showed higher IC_50_ values than ACTH (Figure 6 and Table 1). Furthermore, higher affinities for ACTH in four omMc4rs were consistent with those studies of Mc4rs and Mc3rs in several fishes, such that ACTH may be the original ligand for the MCRs [13,14,15,16,18,19,34,37,66,69,70,71]. Regarding cAMP signaling, omMc4ra1 and omMc4ra2 had higher EC_50_ values, whereas omMc4rb1 and omMc4rb2 showed lower EC_50_ values in response to α-MSH and ACTH (Figure 7 and Table 2). Different pharmacological characteristics between omMc4ras and omMc4rbs further supported that trout Mc4rs might have undergone divergent evolutionary processes.

The spontaneously active conformation of GPCRs in the absence of the agonist results in the constitutive activation of receptors [72]. High basal activities are observed in numerous fish Mc4rs [13,14,15,16,18,19,33,34]. In the present study, of the four omMc4rs, two (omMc4ra2 and omMc4rb1) had high basal cAMP levels, whereas the other two had modest basal activities (Figure 8A). The differences in constitutive activity between these omMc4rs and hMC4R might be attributed to their evolutionarily distinct structures. Although TMDs of MC4Rs are conserved from teleosts to mammals, less homology is present at N-termini and extracellular loops of the four omMc4rs and other teleost Mc4rs compared to hMC4R. N-terminus and extracellular loops play key roles in the modulation of constitutive activities in hMC4R [30,73], as well as luteinizing hormone receptor [74] and thyroid-stimulating hormone receptor [75,76]. However, more research is needed to determine whether these domains contribute to the constitutive activation of the omMc4rs and whether high basal activity of Mc4r is more common among teleosts.

Numerous constitutively active mutants have been reported in several GPCRs, resulting in various human diseases, and inverse agonism will be vital for correcting the diseases caused by activating mutations in GPCRs [32,77]. The constitutive activity of teleost Mc4r may have important physiological functions. Our data showed that AgRP (83–132), an inverse agonist, reduced the basal cAMP levels of four omMc4rs (Figure 8), similar to results on spotted scat, grass carp, and Nile tilapia Mc4rs [17,51]. Although fish have four endogenous antagonists, including agouti signaling peptide 1 (Asip1), Asip2, Agrp1, and Agrp2, Agrp1 shows high identity with tetrapod AgRP at the C-terminus with ten conserved cysteine residues (forming five disulfide bridges) and a highly conserved Arg-Phe-Phe (RFF) motif, which is essential for maintaining the structure and biological function of AgRP [20,78]. Thus, the active fragment in the C-terminal human AgRP behaves as an inverse agonist in fish Mc4rs, indicating that AgRP-induced inverse agonism on MC4R is conserved in vertebrates, and inverse agonism and antagonism of teleost Mc4r might provide new insight for promoting feed intake and growth performance in aquaculture.

Three small molecule inverse agonists of hMC4R were used in the current study to determine their potential modulation on cAMP and ERK1/2 signaling pathways of four omMc4rs (Figure 8). MCL0020 is a neutral antagonist of hMC4R [51,56,79]. MCL0020 replaces NDP-MSH binding to human, grass carp (ciMc4r), and spotted scat (saMc4r) MC4Rs, it and antagonizes NDP-MSH-induced cAMP generation and decreases the basal cAMP production of ciMc4r, whereas it has no effect on basal and ligand-induced cAMP levels on saMc4r [51]. Our results indicated that MCL0020 only bound to omMc4rb1 and hMC4R by displacing NDP-MSH, suggesting that binding sites for this compound and NDP-MSH were overlapping in hMC4R and omMc4rb1, which is different than omMc4ra1, omMc4ra2, and omMc4rb2 (Figure 6). For signaling, MCL0020 had no significant effects on the cAMP level of four trout Mc4rs (Figure 8).

ML00253764 and Ipsen 5i were shown to decrease the basal activities of WT and constitutively active MC4R mutants [48,51,57,58,80]. Both ML00253764 and Ipsen 5i bind to hMC4R orthosterically [15,51]. Results of the present study showed that neither ML00253764 nor Ipsen 5i bound to four trout Mc4rs orthosterically by displacing radiolabeled NDP-MSH (Figure 6), similar to saMc4r, ciMc4r, and ricefield eel Mc4r, indicating that binding sites of these compounds are distinct between fish Mc4r and hMC4R [15,51]. Additionally, both compounds did not cause significant effects on the cAMP levels of four omMc4rs (Figure 8), similar to the results with saMc4r and ciMc4r [51], whereas ML00253784 could decrease basal cAMP generation in ricefield eel Mc4r [15]. 

MC4R-mediated ERK1/2 signaling has a crucial role in the modulation of energy homeostasis [46]. Although teleost Mc4rs have much higher basal activities in Gs-cAMP signaling, constitutive activation of ERK1/2 signaling is only observed in topmouth culter Mc4r but not in saMc4r and ciMc4r [51]. Our results showed that only omMc4rb2 was constitutively active in ERK1/2 signaling, and the other three omMc4rs had similar pERK1/2 levels as the empty vector (Figure 9). For ligand-induced ERK1/2 signaling, α-MSH was shown to stimulate pERK1/2 of four omMc4rs (Figure 9). AgRP, as a biased ligand, was shown to decrease Gs-cAMP signaling and activate ERK1/2 signaling on hMC4R and fish Mc4rs (saMc4r and ciMc4r) [48,51,81]. Our data indicated that AgRP could decrease cAMP levels of four omMc4rs, whereas it significantly stimulated ERK1/2 signaling at omMc4ra1 and omMc4rb1 but not at omMc4ra2 and omMc4rb2 (Figure 9), suggesting that AgRP is a biased ligand for omMc4ra1 and omMc4rb1. Although the physiological roles of fish Mc4r-regulated ERK1/2 signaling are not well understood, our findings presented vital new knowledge on the complexity of fish Mc4r pharmacology.

Three small molecules (Ipsen 5i, ML00253764, and MCL0020) were suggested to act as biased ligands for hMC4R and fish Mc4rs (spotted scat and grass carp Mc4r) with preferred activation on ERK1/2 signaling [48,51,81]. Our data documented that these three compounds only activated ERK1/2 signaling of omMc4rb1 but not the other three omMc4rs (Figure 9). Furthermore, ML00253764 could stimulate cAMP signaling but had no effect on ERK1/2 signaling of omMc4rb2 (Figure 8 and Figure 9). Collectively, these findings suggested that Ipsen 5i, ML00253764, and MCL0020 were biased allosteric modulators for omMc4rb1 with selective activation on ERK1/2 signaling, while ML00253764 behaved as an allosteric agonist for omMc4rb2, preferentially activating Gs-cAMP signaling. Thus, small molecules developed for hMC4R had various effects on the Mc4r in different fishes. We need to investigate the pharmacological characteristics of Mc4r in the intended fish before field trials.

## 5. Conclusions

In summary, four trout *mc4r* genes were located at different chromosomes, and these four Mc4r isoforms showed different pharmacological characteristics, indicating that trout *mc4r* may have undergone different evolutionary processes and have various physiological functions. Trout Mc4rs had constitutive activities in cAMP signaling that were decreased by AgRP. Several MC4R ligands possessed divergent effects on trout Mc4rs, resulting in biased cAMP and ERK1/2 signaling. These findings contributed to a better understanding of pharmacology in Mc4r isoforms encoded by different *mc4r* paralogs. A very recent report investigated the modulation of trout Mc4rb1 by Mrap2a [82]. Taken together, these results will aid further investigations of the physiological role of fish Mc4r in energy homeostasis and of the evolution of MC4R in vertebrates.

## Figures and Tables

**Figure 1 biomolecules-13-01248-f001:**
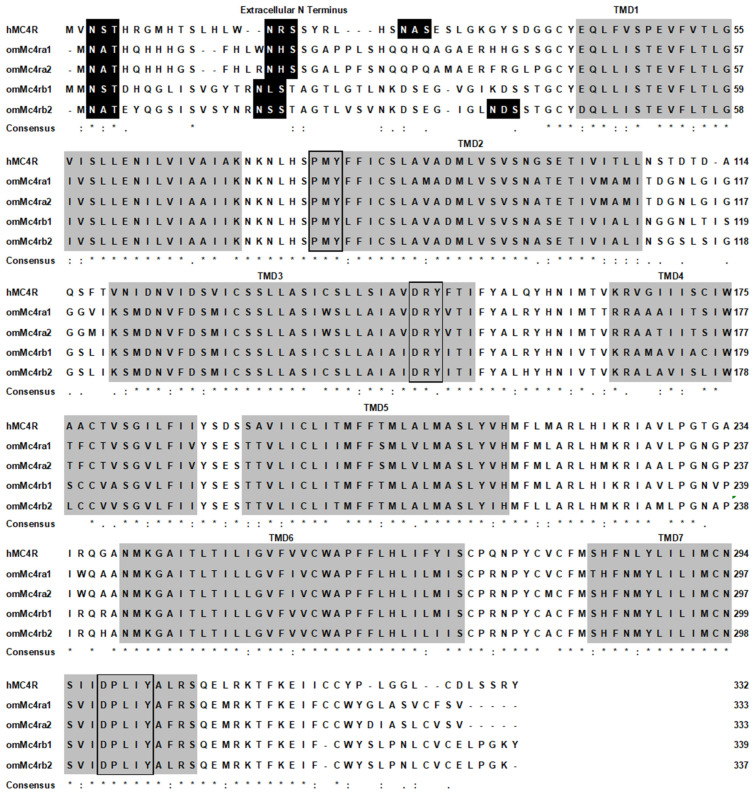
Comparison of amino acid sequences between omMc4rs and hMC4R. Dark boxes indicated potential *N*-linked glycosylation sites. Shaded boxes show putative TMD 1–7. Open boxes show the conserved motifs (PMY, DRY, and DPxxY). Asterisk (*) indicates identical residues (fully conserved residue); colons (:) and period (.) indicate highly conserved and semi-conserved amino acids, respectively.

**Figure 2 biomolecules-13-01248-f002:**
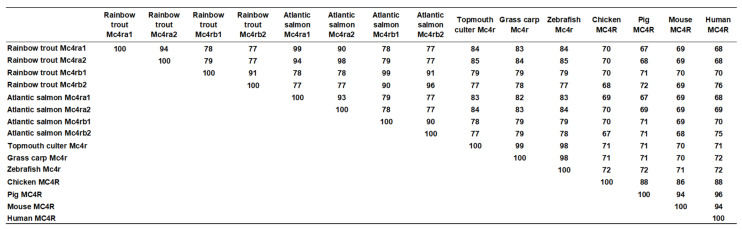
Amino acid sequence identities of omMc4rs and other MC4Rs.

**Figure 3 biomolecules-13-01248-f003:**
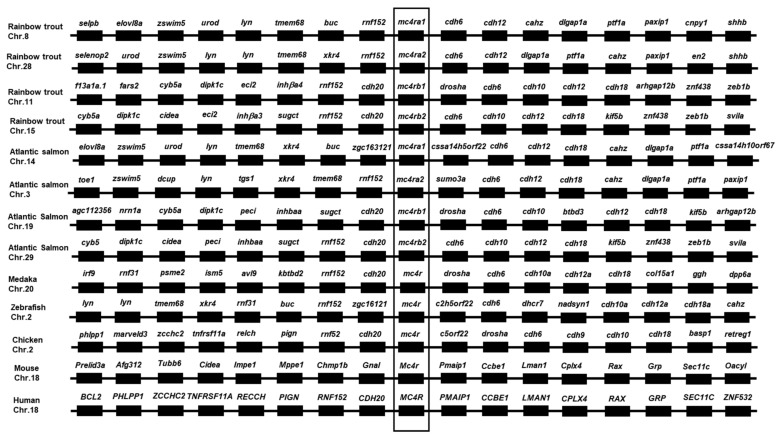
Synteny analyses of MC4Rs. The syntenic genes are displayed as dark boxes linked by lines. MC4R genes are boxed.

**Figure 4 biomolecules-13-01248-f004:**
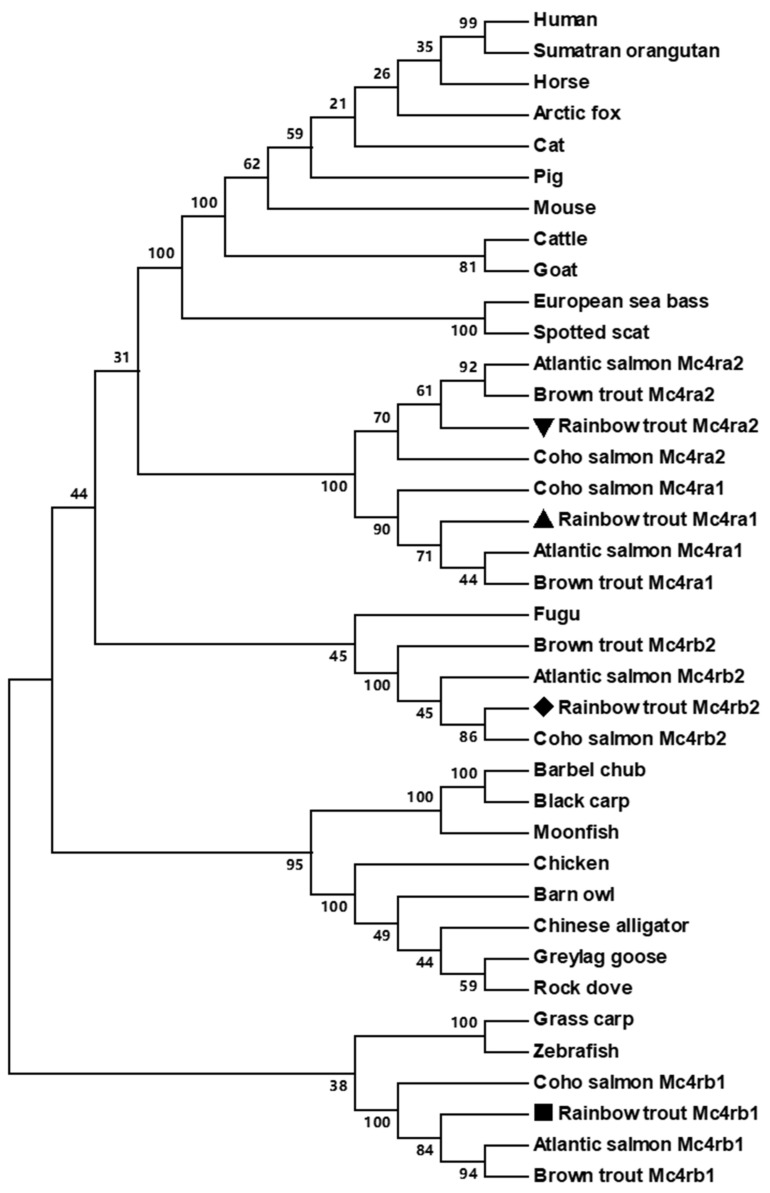
Phylogenetic tree of MC4Rs. The tree was constructed using the neighbor-joining (NJ) method. Numbers at nodes indicate the bootstrap values, as percentages, obtained for 1000 replicates. Shapes indicate trout Mc4r isoforms.

**Figure 5 biomolecules-13-01248-f005:**
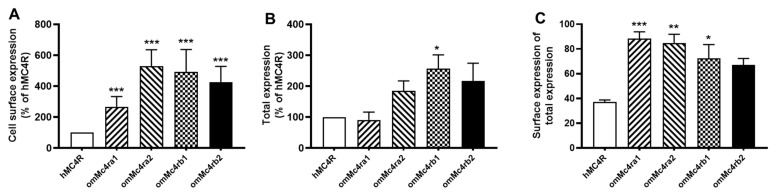
Cell-surface and total expression of trout Mc4rs. (**A**) cell-surface expression; (**B**) total expression; (**C**) surface ratio (surface expression/total expression). HEK293T cells were transiently transfected with omMc4ra1, omMc4ra2, omMc4rb1, omMc4rb2, or hMC4R, as described in Materials and Methods. Data are mean ± SEM from three or four independent experiments. * Significantly different from hMC4R (* *p* < 0.05, ** *p* < 0.01, *** *p* < 0.001).

**Figure 6 biomolecules-13-01248-f006:**
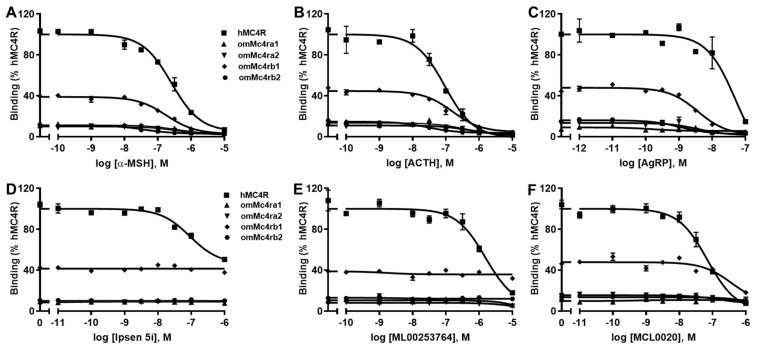
Ligand-binding properties of omMc4rs. HEK293T cells were transiently transfected with hMC4R or omMc4r plasmids, and the binding properties were measured 48 h later by displacing the binding of ^125^I-NDP-MSH using different concentrations of unlabeled α-MSH (**A**), ACTH (1–24) (**B**), AgRP (83–132) (**C**), Ipsen 5i (**D**), ML00253764 (**E**), or MCL0020 (**F**). Data are expressed as % of hMC4R binding ± range from duplicate measurements within one experiment. The curves are representative of at least three independent experiments.

**Figure 7 biomolecules-13-01248-f007:**
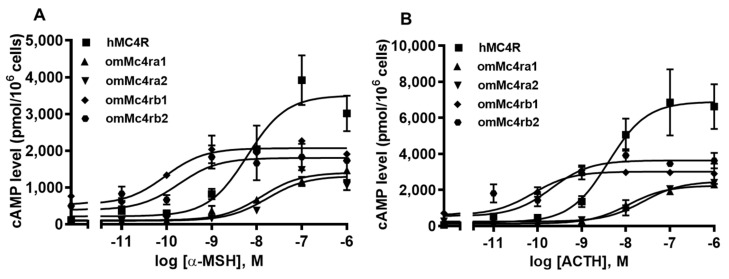
Signaling properties of omMc4rs. HEK293T cells were transiently transfected with omMc4r or hMC4R plasmids, and intracellular cAMP levels were measured by RIA after stimulation with different concentrations of α-MSH (**A**) or ACTH (1–24) (**B**). Data are mean ± SEM from triplicate measurements within one experiment. All experiments were performed at least three times independently.

**Figure 8 biomolecules-13-01248-f008:**
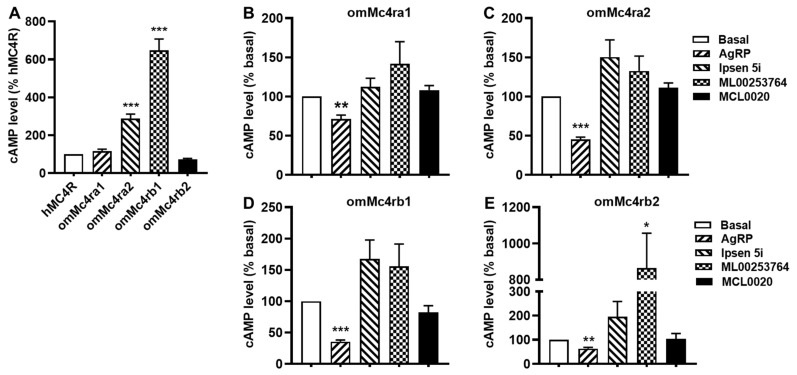
Effects of four ligands on the basal activities of omMc4rs. HEK293T cells were transiently transfected with four omMc4rs plasmids, and intracellular cAMP levels were measured 48 h later by RIA after stimulation without (**A**) or with 10 nM AgRP, 1 μM MCL0020, 1 μM ML00253764, or 1 μM Ipsen 5i at the four omMc4r isoforms (**B**), omMc4ra1; (**C**), omMc4ra2; (**D**), omMc4rb1; and (**E**), omMc4rb2 as described in Section 2. Data are mean ± SEM from three independent experiments. * Significantly different from basal activity of hMC4R or corresponding omMc4r (* *p* < 0.05, ** *p* < 0.01, and *** *p* < 0.001).

**Figure 9 biomolecules-13-01248-f009:**
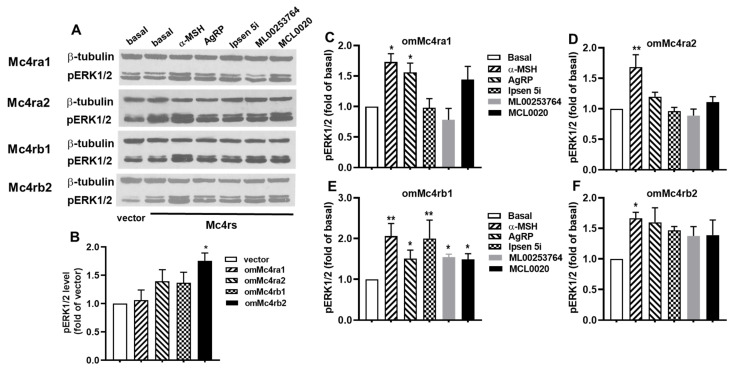
ERK1/2 signaling properties of trout Mc4rs. (**A**) Immunoblots of pERK1/2 in cells expressing omMc4ra1, omMc4ra2, omMc4rb1, or omMc4rb2. HEK293T cells were transiently transfected with omMc4ra1, omMc4ra2, omMc4rb1, omMc4rb2, or empty vector plasmids and were starved overnight 24 h after transfection. Cells were treated with vehicle only or with 10 nM AgRP, 1 μM Ipsen 5i, 1 μM ML00253764, or 1 μM MCL0020 for 5 min. Western blots were performed as described in Materials and Methods. (**B**) Densitometry results of basal pERK1/2 levels of omMc4ra1, omMc4ra2, omMc4rb1, and omMc4rb2. (**C**) Densitometry results of pERK1/2 level of omMc4ra1. (**D**) Densitometry results of pERK1/2 level of omMc4ra2. (**E**) Densitometry results of pERK1/2 level of omMc4rb1. (**F**) Densitometry results of pERK1/2 level of omMc4rb2. Data are mean ± SEM from at least three independent experiments. * Significantly different from basal or vector activity (* *p* < 0.05, and ** *p* < 0.01).

**Table 1 biomolecules-13-01248-t001:** The ligand-binding properties of omMc4rs to different ligands.

		hMC4R	omMc4ra1	omMc4ra2	omMc4rb1	omMc4rb2
B_max_ (%)		100	15.51 ± 3.6^2 c^	10.80 ± 0.49 ^c^	39.66 ± 2.06 ^c^	14.32 ± 1.25 ^c^
α-MSH	IC_50_ (nM)	306.69 ± 45.40	413.39 ± 64.70	384.86 ± 100.62	222.50 ± 42.74	64.20 ± 22.25 ^b^
ACTH (1–24)	IC_50_ (nM)	114.70 ± 27.56	260.83 ± 61.91	375.67 ± 88.27 ^a^	100.46 ± 20.98	22.95 ± 5.62 ^a^
AgRP	IC_50_ (nM)	32.48 ± 7.76	0.42 ± 0.15 ^a^	2.87 ± 1.07 ^a^	3.28 ± 0.63 ^a^	1.54 ± 0.65 ^a^
Ipsen 5i	IC_50_ (nM)	399.86 ± 122.11	N/A	N/A	N/A	N/A
ML00253764	IC_50_ (nM)	1644.57 ± 215.24	N/A	N/A	N/A	N/A
MCL0020	IC_50_ (nM)	62.33 ± 4.82	N/A	N/A	305.05 ± 66.19 ^a^	N/A

Values are expressed as the mean ± SEM of at least three independent experiments. ^a^ Significant difference from the parameter of hMC4R, *p* < 0.05. ^b^ Significant difference from the parameter of hMC4R, *p* < 0.01. ^c^ Significant difference from the parameter of hMC4R, *p* < 0.001. N/A: could not be determined.

**Table 2 biomolecules-13-01248-t002:** The signaling properties of omMc4rs.

	α-MSH		ACTH (1–24)	
	EC_50_ (nM)	R_max_ (%)	EC_50_ (nM)	R_max_ (%)
hMC4R	4.34 ± 1.04	100	4.19 ± 0.60	100
omMc4ra1	11.71 ± 2.59 ^a^	43.78 ± 4.16 ^c^	13.54 ± 2.60 ^b^	48.64 ± 4.32 ^c^
omMc4ra2	24.51 ± 2.26 ^c^	48.30 ± 3.82 ^c^	25.66 ± 2.36 ^c^	48.47 ± 4.56 ^c^
omMc4rb1	0.04 ± 0.02 ^a^	54.63 ± 4.94 ^c^	0.07 ± 0.02 ^c^	48.52 ± 2.40 ^c^
omMc4rb2	0.32 ± 0.07 ^a^	54.15 ± 3.76 ^c^	0.13 ± 0.03 ^c^	53.86 ± 1.42 ^c^

Values are expressed as the mean ± SEM of at least three independent experiments. ^a^ Significant difference from the parameter of hMC4R, *p* < 0.05. ^b^ Significant difference from the parameter of hMC4R, *p* < 0.01. ^c^ Significant difference from the parameter of hMC4R, *p* < 0.001.

## Data Availability

The raw data supporting the conclusions of this article will be made available by the authors upon request, without undue reservation.

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
