# Peer review of "Divergent Pharmacology and Biased Signaling of the Four Melanocortin-4 Receptor Isoforms in Rainbow Trout (Oncorhynchus mykiss)"

_biomolecules, 2023, doi:10.3390/biom13081248_

Round 1

Reviewer 1 Report

The authors present an interesting study of the four rainbow trout MC4Rs with binding affinity and activity. The paper is well written. It would be more fluent when the different introductory paragraphs will be linked. 

I have some questions about the results:

L228: do the authors have a control for the transfection efficiency of each construct: the efficiency should be the same to compare the results, for example by co-transfecting a second plasmid and control that the expression of this second plasmid is the same for all the tested plasmids. 

L230: The authors could repeat rapidly which cell line was used to introduce the assay or in the legend of figure 5.

L234: Why was beta-MSH not used as it is the usual ligand of mammalian MC4R?

Figure 6: why is the binding in percent of hMC4R? and not the percent of labelled NDP-MSH

Table 2: the value of the cAMP basal activity is shown in figure 8a and Table 2. In Figure 7, it looks like it is omMC4Ra1 that has the higher basal activity.

The paper could have more impacts if the MRAP2 action would be tested on the different four omMC4Rs since it is known since 2014 see Dores et al. (Molecular evolution of GPCRs: Melanocortin/melanocortin receptors R. M. Dores, R. L. Londraville, J. Prokop, P. Davis, N. Dewey and N. Lesinski J Mol Endocrinol 2014 Vol. 52 Issue 3 Pages T29-42).

There are minor mistakes

Author Response

Reviewer #1

  1. It would be more fluent when the different introductory paragraphs will be linked. 

Reply: I respectfully disagree. Long paragraphs are difficult for readers because they do not give readers space to breath. A good idea is to have a paragraph less than half page in double-spaced manuscript. When reviewing manuscripts,  I frequently ask authors to break long paragraphs into shorter ones to give the reader space to breathe.

  1. L228: do the authors have a control for the transfection efficiency of each construct: the efficiency should be the same to compare the results, for example by co-transfecting a second plasmid and control that the expression of this second plasmid is the same for all the tested plasmids. 

Reply: Thank you for your valuable suggestions. We did not include a control for transfection efficiency for each construct in our study. In co-transfection experiments, when a cell picks up one plasmid, it will always pick up the second plasmid. In our experiments, equal constructs were transfected into the cells. Additionally, all gene sequences were cloned into the same empty vector (pcDNA3.1), and all constructs, including the four trout Mc4rs and human MC4R, have identical vector sizes. Each study was independently performed at least three times to ensure reliability. Furthermore, human MC4R was chosen as a control for comparative purposes.

  1. L230: The authors could repeat rapidly which cell line was used to introduce the assay or in the legend of figure 5.

Reply: Done. HEK293T was used in all studies.

  1. L234: Why was beta-MSH not used as it is the usual ligand of mammalian MC4R?

Reply: a-MSH is highly conserved in vertebrates, and trout a-Msh exhibited high identities (100% or 92%) with human a-MSH (two pomc genes in trout). However, trout b-Msh only shared 78% or 68% identities with human b-MSH. We have added this information in supplemental files.

  1. Figure 6: why is the binding in percent of hMC4R? and not the percent of labelled NDP-MSH

Table 2: the value of the cAMP basal activity is shown in figure 8a and Table 2. In Figure 7, it looks like it is omMC4Ra1 that has the higher basal activity.

 Reply: 1. The binding in percent of hMC4R was used instead of the percent of labeled NDP-MSH because hMC4R was employed as a control for comparison in our study. During the binding assay, the two-well plate cells were solely incubated with labeled NDP-MSH without any other unlabeled ligands, allowing us to measure the maximal binding values (Bmax, a pharmacology parameter). To calculate the binding value for each trout Mc4r, we divided the Bmax of hMC4R. This data transformation had no effect on determining the IC50 for each construct.

  1. 2. We have removed the basal activities from Table 2 and retained this data solely in Figure 8.

  1. After thorough examination of the original figures, we discovered an error in Figure 7A. The correct labeling should be as follows: omMc4ra1 should be labeled as omMc4rb2, omMc4ra2 should be labeled as omMc4rb1, omMc4rb1 should be labeled as omMc4ra2, and omMc4rb2 should be labeled as omMc4ra1. We have addressed this issue and made the necessary corrections in Figure 7A to ensure accurate representation of the data.

  1. The paper could have more impacts if the MRAP2 action would be tested on the different four omMC4Rs since it is known since 2014 see Dores et al. (Molecular evolution of GPCRs: Melanocortin/melanocortin receptorsR. M. Dores, R. L. Londraville, J. Prokop, P. Davis, N. Dewey and N. Lesinski J Mol Endocrinol 2014 Vol. 52 Issue 3 Pages T29-42).

Reply: Thank you for your valuable suggestion. We agree that testing Mrap action on the four different omMc4rs, as mentioned in Dores et al. (2014), could significantly impact our research. Our lab specializes in MC3R/MC4R pharmacology and regulation by MRAPs in various species, and we have published several papers in this area. During genomic study of trout, we identified four mrap2 genes and one mrap1 gene. Investigating the Mrap-regulated Mc4r pharmacology in trout has been a substantial endeavor, and we have additional papers specifically dedicated to this topic.

Reviewer 2 Report

This manuscript analyzed the pharmacological characteristics of four trout Mc4r isoforms (omMc4rs), using a-MSH and AgRP, MCL0020, ML00253764, and Ipsen 5i as MC4R inverse agonists. The results showed the presence of biased allosteric modulation of trout Mc4r isoforms. The findings shall facilitate the investigation of potential physiological functions of the four mc4r paralogs. While this study was fairly well executed, with methodology well established in the authors’ laboratory, the manuscript could be further improved:

1.     It is hard to understand how the cell surface and total expression of Mc4rs were determined, although the authors have provided two references (34, 64). More details should be given about this set of experiment. Did the cell surface expression of these Mc4rs change upon addition of Mc4r agonists? Have you performed western analysis of Mc4rs by fractionation of membrane proteins? Or immunofluorescence staining with flag-tagged Mc4r isoforms?

2.     It is interesting that omMc4rb1 and omMc4rb2 respond differently to the inverse agonist such as ML00253764. Is this related to the cell face/total expression ratio of these two isoforms? There is more cell surface Mc4rb2 expressed on the cell surface, but the total Mc4rb2 is not higher than hMc4r. Was the differential response of Mc4rb1/b2 related to the intracellular level/domain of these isoforms? Have you tried to analyze the potential binding site by docking?

1.     There is frequent switch of active and passive voices in the writing. It is better to be consistent and use active voice.

Author Response

Reviewer #2

  1. It is hard to understand how the cell surface and total expression of Mc4rs were determined, although the authors have provided two references (34, 64). More details should be given about this set of experiment. Did the cell surface expression of these Mc4rs change upon addition of Mc4r agonists? Have you performed western analysis of Mc4rs by fractionation of membrane proteins? Or immunofluorescence staining with flag-tagged Mc4r isoforms?

Reply: Thank you for your valuable suggestions. We have updated the Materials and Methods (Line 135-146) section to provide more details about how we determined the cell surface and total expression of Mc4rs. In this study, we assessed cell surface/total expression without incubating ligands and did not investigate changes in cell surface expression upon addition of Mc4r agonists. As for the western analysis of Mc4rs, we have not performed fractionation of membrane proteins for this purpose. In previous studies and our unpublished data, MC4R blots are very messy, with numerous bands. See also Kathy Mountjoy’s publications.  Therefore, it is very  challenging to detect MC4R expression using western blotting. Additionally, we have not conducted immunofluorescence staining with flag-tagged Mc4r isoforms in this study because it provides only qualitative data. Flow cytometry provided quantitative data.

  1. It is interesting that omMc4rb1 and omMc4rb2 respond differently to the inverse agonist such as ML00253764. Is this related to the cell face/total expression ratio of these two isoforms? There is more cell surface Mc4rb2 expressed on the cell surface, but the total Mc4rb2 is not higher than hMc4r. Was the differential response of Mc4rb1/b2 related to the intracellular level/domain of these isoforms? Have you tried to analyze the potential binding site by docking?

Reply:  Previously, we showed that prolonged incubation (for example, 24 hours) of ML00253764 can induce intracellular hMC4R to traffic to the cell surface by serving as a pharmacological chaperone. Short-term treatment with ML00253764 revealed its inverse agonist activity in cAMP assays. We do not have a molecular understanding of the differential response of omMc4rb1/b2 to ML00253764. Chimera and reciprocal mutations are needed to clarify this.

The differential response of Mc4rb1/b2 may be attributed to the presence of different binding sites or distinct activation mechanisms. Further research is required to pinpoint their precise binding sites (as you mentioned) and to elucidate their activation mechanisms, using chimeras and reciprocal mutants. Utilizing AI tools like AlphaFold to predict potential binding sites is a promising approach in this regard.

  1. There is frequent switch of active and passive voices in the writing. It is better to be consistent and use active voice.

Reply: Thanks for your kind suggestions. We have carefully went through the manuscript with your comment in mind to make sure the appropriate voice was used.